# Effective Implementation of Integrated Area Development Based on Consumer Attractiveness Assessment

Ilya Zelenskiy [1], Danila Parygin [1,*] , Oksana Savina [1], Alexey Finogeev [2] and Alexander Gurtyakov [1]

[1]  Department of Digital Technologies for Urban Studies, Architecture and Civil Engineering, Volgograd State Technical University, 1 Akademicheskaya Str., Volgograd 400074, Russia

[2]  Department of Computer-Aided Design Systems, Penza State University, 40 Krasnaya Str., Penza 440026, Russia

*   Correspondence: dparygin@gmail.com

**Abstract:** This article examines the 11th UN Sustainable Development Goal through the lens of the needs of citizens. The study proposes to compare the means and goals of integrated area development (IAD) and sustainable development. It was proposed to implement a decision support system for choosing options for the area development, based on the assessment of indicators for achieving the desired result. The purpose of the article is to consider the possibility of using the attractiveness of the territory for potential consumers as such an indicator. The calculation method, as well as methods for collecting, filtering, and processing open sources of data on the state of the urban environment, were developed to evaluate this indicator. The obtained methods were tested on a sample of apartments in the Volgograd, Russia, in order to verify the adequacy of the proposed indicator. A comparison of the dynamics of the resulting assessments of attractiveness and the market prices of the studied apartments showed a correlation and adequacy of the proposed methods for practical application. The main limitation remains the need for large-scale collection and preliminary processing of data for the assessment, as well as setting up of the assessment method configuration: criteria, scales, etc.

**Keywords:** integrated area development; sustainable development; real estate; consumer attractiveness assessment; infrastructure provision assessment; natural language processing; decision support system

## 1. Introduction

The well-being of mankind today is increasingly determined by the quality of life in cities. However, the rapid growth of cities leads to a sharp aggravation of a wide range of vital problems [1].

The structure of modern cities is characterized by a complex spatial distribution of people and activities, as well as uneven infrastructure coverage [1]. Creating a comfortable urban environment for human life activity is becoming one of the priorities in the development of cities. It is necessary to work on the efficient use of urban areas to solve this problem. A relevant topic within the framework of this issue all over the world is the reorganization of inefficiently used urban areas, for example, former industrial zones, which have long worked out their service life and are morally obsolete [2].

The development of industry in Europe was accompanied by urbanization and the formation of centers of intensive development based on large cities. The largest cities in Europe, primarily capital cities, at a certain stage of their development turned out to be a place of concentration of industrial enterprises of various industries that had a negative impact on the environment [3].

For example, the city of Volgograd, Russia, during the 19th–20th centuries, went through active phases of chaotic industrial development, characterized by all the accompanying problems. It began with the development of a transit railway hub and the oil sector, at the origins of which were the Nobel brothers back in the time of Tsaritsyn (the first name of the city of Volgograd during the Russian Empire). It continued further during the period

of industrialization of Stalingrad (the name of Volgograd during the USSR period), when extended coastal territories along the Volga River were given for the development of the tractor, chemical, metallurgical and other industries. Economic growth was reflected in the organization of urban life in stretching the city in a fairly narrow strip of more than 60 km along the river. At the same time, residential areas turned out to be a chain of islands of urbanization periodically completely cut off from the river by factory infrastructure [4].

As a result, urban proportions were violated, and hypertrophied traffic flows were generated, which provoked social tension [3]. The development of all cities that reach the post-industrial level is characterized by the need to solve such problems of reorganizing industrial territories, improvement and the renovation of old districts, developing unused areas, etc.

Territorial development is described in the literature and legislation of different countries as a process of change in which the exploitation of natural resources, the redistribution of investments, the orientation of scientific and technological development, the development of the individual, and institutional changes are coordinated with each other and strengthen the current and future capacity to meet human needs [5]. The person responsible for carrying out activities within the framework of the integrated area development (IAD) needs to prepare a number of documents at the stage of preparing and approving the layout of the developed area [6]. In particular, it is necessary to prepare a plan for future development that meets the requirements of sustainable development, permitted uses of land and real estate, limiting development parameters and other similar restrictions [7]. Such integrated urban planning faces the challenges of making effective and informed management decisions [8]:

- which of the possible development options will be more preferable;
- how the implementation of each of them will affect the developed area;
- how it will affect the quality of life of the population, etc.

At the moment, there is no universal tool that could be used in the analysis of the urban environment to support decision-making and evaluate development options within the framework of IAD [9], although work in this direction is actively underway [10]. It would be useful for decision-makers to be able to automatically generate the optimal development option or options, taking into account external constraints. However, the solution of such a problem requires, first of all, the definition of optimality criteria, i.e., some measurable quantitative "quality indicators" of development options. For example, the following characteristics of the territory can be used as indicators:

- transport accessibility, i.e., the opportunity to use transport infrastructure facilities and transport services for various groups of the population;
- the provision of social infrastructure, i.e., an indicator of satisfaction of the population with the existing objects of social infrastructure;
- consumer attractiveness of the real estate, i.e., an indicator of the ability of the territory to attract a potential consumer (resident, tenant, entrepreneur, etc.);
- etc.

The study in this article aims to:

1. Consider the use of the final consumer attractiveness of real estate in an area being developed as a criterion for the optimality of development options.
2. Check the adequacy of consumer attractiveness for practical application in IAD.

## 2. Background

Redevelopment is a process of solving the problems of transforming existing urban areas in order to use those more efficiently to meet the needs of stakeholders (residents, landowners, etc.) [11]. This term is usually understood as any new construction on a site that has pre-existing uses. It involves the development of land use to revitalize the physical, economic and social structure of urban space [12].

The Urban Renewal Program, conducted by the US government throughout almost the entire 20th century, can serve as an example of this process [12]. The essence of urban renewal was to clean up decayed urban areas (former industrial estates, abandoned buildings, depressed areas, slums, etc.) and create opportunities for the construction of higher-end housing, new businesses and other objects that can derive more practical benefits from specific urban areas. The main purpose of the Urban Renewal Program was to restore economic viability of a given city area by attracting additional private and public investments and by stimulations business start-ups and survival in that area [12].

First, area redevelopment projects in the US within the urban renewal were mainly focusing slum clearance with their primary goal being the elimination of poor housing conditions and building less crowded and cleaner public housing [13]. Later, projects have included, for example, the redevelopment of large sections of New York city and New York state by Robert Moses, who also directed many such projects as new bridges, highways and public parks construction [14].

The same process, but with the name of urban regeneration started in UK significantly earlier, in the 1850s, when the condition of London slums began to attract the attention of various reformers and social philanthropists. The first area to be the subject of redevelopment was 'the Devil's Acre' near Westminster, a well-known slum of its time. Further slum eviction began with the Rochester Buildings, which were later known as Blocks from A to D of the Old Perkin's Rents Estate. Then, an experimental social housing estate was founded [15]. By the time of 1869, the Peabody Trust built one of its first housing estates at Brewer's Green, between Victoria Street and St. James's Park [16]. After the Cross Act of 1875, the last remaining of the Devil's Acre were eliminated and further Peabody estates were built [16]. By the time of 1882, the Peabody Trust began subsequently repeating the same of their designs in various housing estates in London [15]. Currently, there are two main urban regeneration projects going on in the British capital, Elephant Park [17] at Elephant and Castle [18] and at Stratford [18].

In the last decade, a similar situation developed in mainland China, where the patterns of urban development were significantly similar to those in many Western countries. It is possible to draw parallels with the US Urban Renewal Program. Older neighborhoods in both countries were seen as outdated and depressive and were encouraging local governments and development interests to cooperate for downtown redevelopment [19]. China's urbanization is also accelerating in the current context of rapid global urbanization, and the rational planning and sustainable use of state land and space have become a growing concern [20].

Basically, the same program, also called urban regeneration, took place in South Korea. The program began with the reconstruction following the Korean War. In 2017, the urban regeneration project proceeded under the new name of New-Deal Urban Regeneration [21].

The redevelopment of urban areas has also been developed on the territory of the Russian Federation [22,23]. An example of this process is the Housing Renovation in Moscow Program, launched in 2017. The main focus of the program was the demolition of morally and physically obsolete Soviet residential buildings [24].

However, redevelopment projects and programs have been incredibly controversial, including the previously mentioned urban renewal [25] and urban regeneration programs [13]. For example, the residents displaced by redevelopment were often undercompensated and some are not compensated at all [26].

The most obvious example is that until 1970 the displaced owners and tenants in the USA received only the government-mandated "just compensation" specified in the USA constitution. This measure of compensation covered only the fair market value of the taken property, which quite often was not the main element of the property's price, and avoided compensating a variety of incidental losses like, for example, moving expenses, loss of favorable financing and most notably, business losses, such as loss of business goodwill [13]. As a result, many neighborhoods were thrown away from their homes with nearly zero ways to reestablish. What was meant to be slums eviction turned out to be

more like slums relocation as removing one depressive area were leading to establishing another, perhaps, the bigger one which further led to increasing social tension and fueled the growing crisis [13]. The same is also fair for the China program. The focus of the projects' attention turned out to be skewed towards building plenty of highways to reach large scale urban sprawl. Therefore, they failed to provide enough support and attention for residents of the areas under redevelopment, who often were the low-income people [19].

Moscow renovation faced similar problems in many respects. The owners were provided with apartments in new houses to replace the demolished one, which was equivalent in terms of characteristics, but in no way equivalent in value [27]. This problem has not bypassed the business, for which the relocation forced due to the demolition of the current location and the direct and indirect costs and risks associated with it sometimes became an existential problem [28].

Urban renewal was forced to acquire new restrictions, conditions and directions of development over time. As a result, it is a mix of renovation, selective demolition, commercial development, and tax incentives which most often used to revitalize urban territories in the US nowadays, due to many controversies generated by urban renewal throughout the years.

Volgograd, whose stages of development are given in the introduction, may face similar situations in the coming years. Firstly, the city was one of the first in Russia after Moscow to launch the Housing Renovation Program, which will obviously become a trigger for new and unexplored processes of such transformation for a non-capital city. Secondly, a number of industrial enterprises located along the river bank were decommissioned. Now, draft competitions for projects for the development of vast spaces in promising areas of Volgograd are actively going on, in anticipation of the launch of integrated development programs for the cleared territories [4,23].

A number of conclusions can be drawn based on the analysis. It is critically important that the creation of an urban environment that is comfortable for a person is also limited by the condition of reducing the negative impacts of economic and other activities on the person himself and on the natural environment. It is necessary to ensure the protection and rational use of natural resources for the benefit of present and future generations. Thus, the development and redevelopment of cities should be subject to the principles of sustainable development [29].

It will be practically impossible to create a comfortable urban environment without adherence to these principles, since a system built without regard to sustainability will inevitably tend to self-destruction [13]. Goal 11 of the "Sustainable cities and communities" in the UN list of Sustainable Development Goals [30] is directly related to the issues of urbanization and the functionality of the urban environment. Here, an integrated approach to solving emerging problems is required, as in any other goal from the list.

A new Chapter 10 "Integrated Area Development" was introduced into the Urban Planning Code of the Russian Federation by Federal Law No. 494-FZ of 30 December 2020 [6]. This chapter regulates a special type of urban planning activity aimed at creating favorable living conditions for citizens, updating the living environment and common areas of settlements, urban districts, developing former industrial zones, etc. [6,7]. However, it cannot be said that the institution of IAD is new for Russia. These activities were carried out within the framework of the process, which received the name of the integrated and sustainable development of areas in the legislation long before the adoption of the relevant federal law [6]. In fact, the binding of this institution to sustainable development with a change in legislation has moved from the name of the institution to the description of its goals. The following IAD goals are assumed by the new edition of the Urban Planning Code of the Russian Federation:

- ensuring a balanced and sustainable development of the city;
- the creation of the necessary conditions for the development of transport, social, engineering infrastructures, improvement of the city's areas;

- increasing the efficiency of using city areas, including the formation of a comfortable urban environment, the creation of service places and places of employment;
- creating conditions for attracting extra-budgetary sources of financing for the renovation of built-up areas.

Integrated should be understood as a balanced development of areas, as follows from the above description of the goals. Therefore, the balance implies the comprehensiveness of areas development, the improvement, for example, of its housing characteristics and all types of infrastructure, the creation of new jobs, etc. [6]. In this case, the integrated in the development of urban areas should be considered a key means or tool for achieving the sustainability of cities. World experience shows [13,24,26,31] that working with the area in one particular direction (for example, only housing development) without regard to the rest negatively affects the stability of the city as a whole [13,32].

The scheme (Figure 1) of the correlation between the redevelopment of urban areas, IAD and the sustainable development of cities was formed, taking into account the analyzed sources.

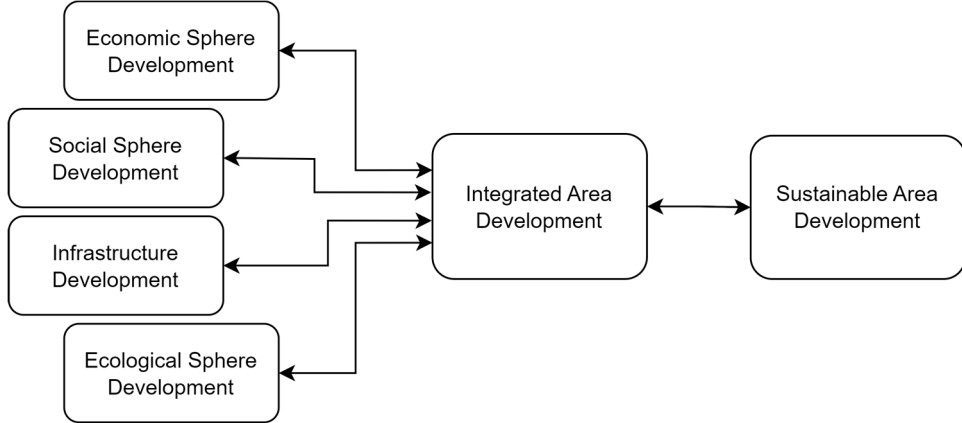

**Figure 1.** Relationship between sustainable development and integrated area development.

Thus, the introduction of the IAD institute into the urban planning legislation is a documentary confirmation of the commitment of the Russian Federation to the 11th goal of sustainable development from the UN list [30].

The relevance of the problem of the IAD and sustainable development is due to modern trends in urban sprawl, changes in the principles of formation and development of settlements, as well as the requirements for urban design of a favorable environment and other factors. The above is related to the commitment of states to the UN sustainable development goals, in particular the already mentioned goal 11 [30]. At the same time, the pace of achieving the sustainability of cities in the world is insufficient to achieve this goal in principle, as the statistics of recent years show [33]. This statement is true for the vast majority of developed countries, including the USA, Great Britain, Germany and France [33].

Based on the foregoing, the problem field of this study was identified and can be described by the following statements:

- the practice of implementing projects for the transformation of areas to correct the mistakes of the initial development has a long-term background, characterized by the successful application of approaches to the reconstruction of physical space and the formation of new user meanings;
- however, successful experience was the result of many negative attempts to transform existing urban planning solutions;
- at the same time, a series of newly created projects continues, which in the future may become new problematic urban areas;

- and neither the global understanding of the "mechanics" of sustainable development, including those expressed in terms of the UN goals, nor the legislation developed in different countries can yet reverse this situation;
- certainly, a radical transformation of cities is a costly process, and economic conditions have a significant impact on the final result of development projects;
- in this regard, it is proposed to check the applicability of the approach to IAD, based on taking into account consumer preferences, as a simultaneous characteristic of the social and economic aspects of the development of cities, which are a space of public interaction of personal interests, for the sustainable development of urbanized areas.

## 3. Methodology

Territory management within the framework of the idea of integrated and sustainable development is always a task related to the monitoring and analysis of spatial data [34]. The effectiveness of management decisions greatly depends on the reliability, relevance and availability of tools for analyzing and visualizing various data sets. This is determined mainly by the specific characteristics of the area: geographical location, social, demographic and economic characteristics, and the presence of natural risks, etc. [34].

The task of information support for area development projects in terms of sustainability are set at all levels from global to local [35]. Nowadays, the effectiveness of their solution significantly depends on the use of modern innovative technologies, in particular, information technologies [36,37]. At the global level, statistical data are collected by countries and regions within the framework of the work of the largest international organizations: the UN, the World Bank, the WHO, etc. [34]. Typically, such data is publicly available and published on the websites of organizations in the format of annual reports [38].

At the local level, the most detailed and up-to-date information about the state of urban areas is contained in geographic information services (GIS). For example, information about objects in these areas is contained in user posts (ads) on specialized sites [12]. Then, the key problem will be the presentation of information in the form of textual descriptions of objects in natural language, i.e., in a wholly or partly unstructured form. In this regard, there is a need to extract structured data from the posts about the characteristics of objects that are significant for analysis [12]. Descriptions of objects must be reduced to a single pattern suitable for machine analysis.

The following areas of activity can be identified as part of the IAD [6]:

- improving the quality of the urban environment;
- the improvement of living conditions of citizens;
- the development and creation of conditions for the development of infrastructure (transport, social, engineering);
- area improvement;
- improving the efficiency of the use of areas;
- creating conditions for attracting extrabudgetary sources of financing.

The following areas are identified as the main subsystems of sustainable development of urban areas [39]:

- economic development;
- infrastructure;
- social sphere;
- natural environment.

In this regard, it is possible to single out a number of links between the subsystems of sustainable urban development and the IAD (see Figure 2).

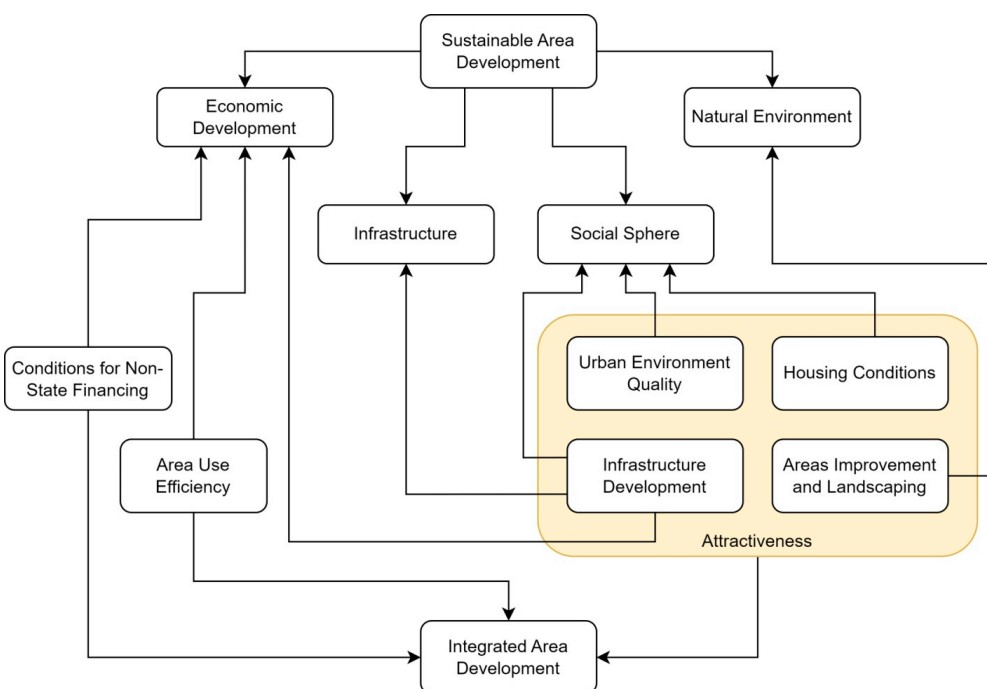

**Figure 2.** Connections of subsystems of sustainable development and integrated area development.

At the same time, uniform and harmonious development in all the listed areas of the IAD will be a condition for achieving sustainability in the development of the city. Accordingly, the problem of decision support should be solved by offering decision-makers the most optimal options for building the area being developed. Then, the assessment of possible options from the standpoint of the listed areas of development can serve as optimality criteria.

The general scheme of the decision support system takes the form shown in the Figure 3 [40]. The system should have blocks for data collection (various parsers, shell modules, etc.), data analysis (this should include blocks for generating and optimizing options for the development of territories) and access (user APIs, etc.).

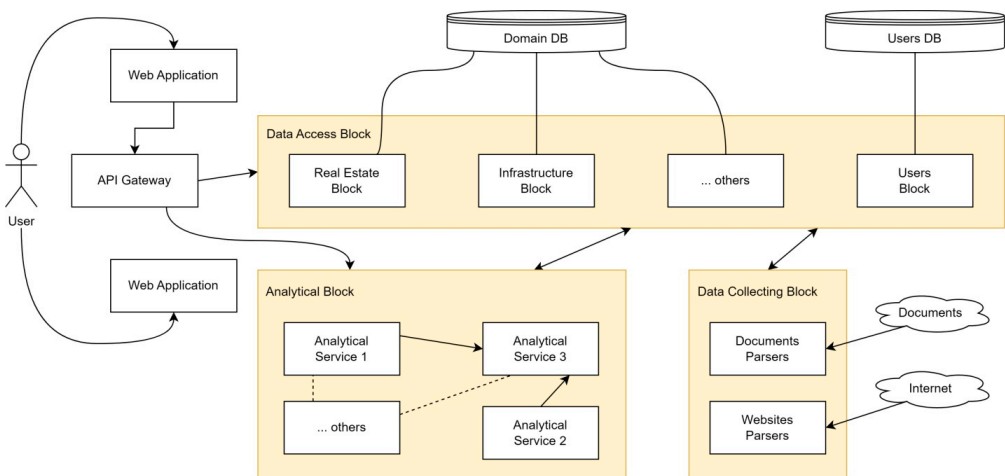

**Figure 3.** Architecture of the developed decision support system.

The task of working out and evaluating various options (scenarios) for the area development in terms of their effectiveness can be solved by various methods, for example, using cognitive modeling [41].

Infrastructure, the quality of the urban environment, housing conditions and area improvement are proposed to be combined into an integral indicator of consumer attractiveness (highlighted in color in Figure 2). Then, it is necessary to solve the problem of choosing the best option with this indicator as an optimality criterion for use in the analytical block of the decision support system. Changes in this indicator as a result of the potential implementation of certain development options will reflect their impact on the overall "comfort" of citizens in developing areas, and, as a result, on potential demand from consumers.

## 4. Results and Discussion

### 4.1. Implementation of the Area Attractiveness Assessment Methodology

The assessment of the area attractiveness is proposed to be carried out through an integral indicator of the attractiveness of the objects located on it. It was necessary to solve the problem of descriptions of individual objects in natural language in user ads in order to calculate their consumer attractiveness. A method for structuring data from user descriptions of objects and supplementing them with information about the surrounding infrastructure needs to be developed for this.

Several standard approaches for extracting information from unstructured text are now widely used [42]:

1. Regular expression approach.
2. The use of classifiers:

   - generative (Naive Bayes classifier);
   - linear classifiers (logistic regression);
   - statistical models (hidden Markov model, conditional Markov model, conditional random field).

The Tomita-parser [43], created by Yandex, was chosen from the currently available set of technologies for the semantic analysis of unstructured text in natural language, such as General Architecture for Text Engineering (GATE) [44], Natural Language Toolkit (NLTK) [45], OpenNLP [46], DeepDive [47], IBM Watson [48], Wolfram Natural Language Understanding (part of the Wolfram Alpha system) [49], etc. One of the main advantages of the Tomita-parser over the other tools listed above, primarily focused on working with the English language, was its focus on working primarily with the Russian language and its specific grammar [43].

As a result of the work, Tomita-parser generated facts, fields where structured data extracted from the text was written. The desired characteristics of real estate acted as facts within the framework of this task [50].

Tomita-parser implements the following data processing order [51]:

- tokenization;
- lexical analysis (recognition of keywords and their combinations by internal dictionaries);
- parsing according to grammars;
- interpretation (output) of results.

The analyzed texts of ads, and the configuration on the basis of which the analysis was carried out, were submitted to the input of the Tomita-parser. The parser provided as output an XML file that contains the parse trees of the input texts suitable for further machine processing.

The corresponding configuration had to be created in order to use the Tomita-parser to solve the problem [50]. The following composition of configuration files has been defined:

- a set of master configuration files (one for each type of the real estate being processed);
- set of formal context-dependent grammars [43] of the form $S \rightarrow S_1 S_2 \ldots S_n$, where S is the non-terminal of the grammar, and $S_1 S_2 \ldots S_n$ is the right-hand side of the rule (terminals and/or non-terminals) that describe the rules for representing the desired characteristics in the text;

- additional keyword dictionaries for individual grammars (for example, for city and district names);
- the root dictionary dic.gzt, which describes the entries for each generated grammar;
- fact file fact_types.proto with a description of the fields of all the facts (characteristics of real estate objects) searched for in the texts.

Its own context-sensitive grammar, which describes the rules for deriving and interpreting chains of words by which this characteristic can be represented in ad texts, was written as part of the task for each desired characteristic.

Tomita-parser has been configured and used as follows:

- the source of texts for processing was the texts of ads supplied to the parser through the system input-output stream;
- the result of the parser's work was presented in the form of an XML tree and returned to the place where the parser was called through the system input-output stream;
- the calling module received the XML parse tree generated by the Tomita-parser and transformed it into structures describing real estate objects in the module's memory.

Information about the address of the actual location was necessarily extracted from the texts of ads for further binding of objects to the area.

A software shell that allows convenient launch of the tool with batch transfer of analyzed information to it via standard system input-output streams bypassing the file system and one-time processing of the results obtained was written. This was carried out in order to improve the usability of the Tomita-parser, which is supplied as an executable file.

The surrounding infrastructure affects the usability of real estate and, as a result, their sustainable [52] ultimate consumer attractiveness. Open data on infrastructure facilities in the city, presented on geoinformation services, were used to assess the surrounding infrastructure.

Google Maps was chosen for the current study as the infrastructure data source [53]. The addresses of the researched real estate objects from user ads, obtained as a result of data structuring, were subjected to geocoding by means of the selected Google service. The coordinates obtained in this way were used as starting points on the map to search for the surrounding infrastructure.

The data on infrastructure facilities presented in the service were divided into types depending on the purpose of the facilities within the city. It was decided to make the assessment separately for each infrastructure facility type at the current research stage. The following infrastructure types were selected to be used in the assessment of infrastructure provision among the defined in the Google Maps:

- education (school);
- healthcare (health);
- public transport (transit_station);
- pharmacies;
- sports (gym);
- hairdressing (hair_care);
- public catering (cafe);
- grocery (grocery_or_supermarket);
- ATMs (atm);
- shopping (shopping_mall).

The possibility of assessing the infrastructure provision was initially considered on the basis of the standards that are legally enshrined in the Russian Federation in the form of building codes and regulations, as well as sanitary standards. Thus, the set of rules 42.13330.2016 establishes specific relationships between the population of an urban area and the areas and capacity, for example, of preschool institutions, which the urban infrastructure must meet to meet the needs of the population. This set of rules establishes similar requirements for general educational organizations, healthcare institutions, leisure and recreation institutions, etc. However, this approach required information that was

not available in open sources, such as information about the actual current workload, for example, of general education institutions.

A simplified methodology for assessing the surrounding infrastructure of real estate objects was developed taking into account the considerations described above. The following formula is proposed:

$$I_i = 10 - mean(D)/100 - p_i, \tag{1}$$

where

$I_i$—final assessment of the object for the $i$ type of real estate;
$D$—list of distances from the estimated object to the found infrastructure objects;
$p_i$—total fine for the $i$ type of real estate.

The scheme of the infrastructure assessment algorithm is shown in Figure 4.

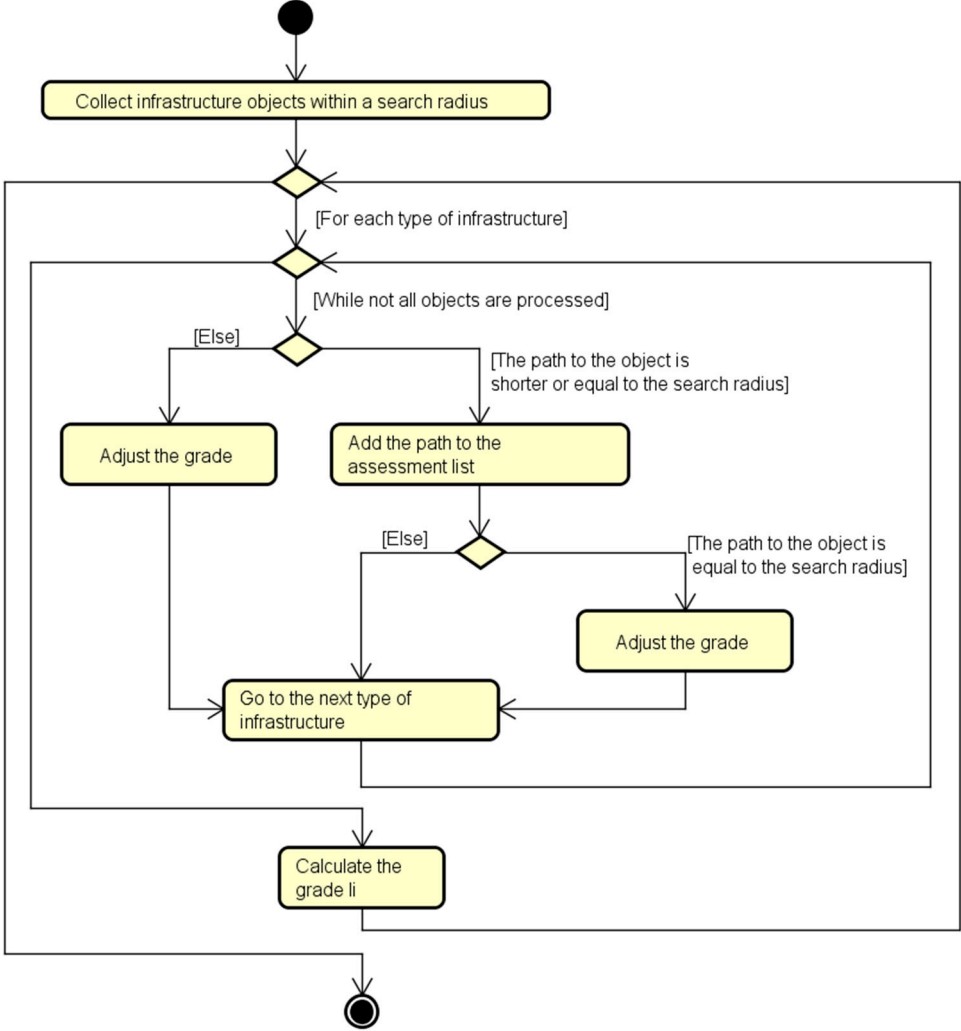

**Figure 4.** Infrastructure assessment algorithm.

The assessment of each object under study in the area under assessment was calculated as the sum of assessments for a variety of criteria. The direct weighted sum method was selected from a variety of multi-criteria assessment methods. The standard adjustment mechanism for real estate valuation was chosen to further account for negative factors. The scheme of a complex algorithm for assessing the consumer attractiveness of real estate objects is shown in Figure 5.

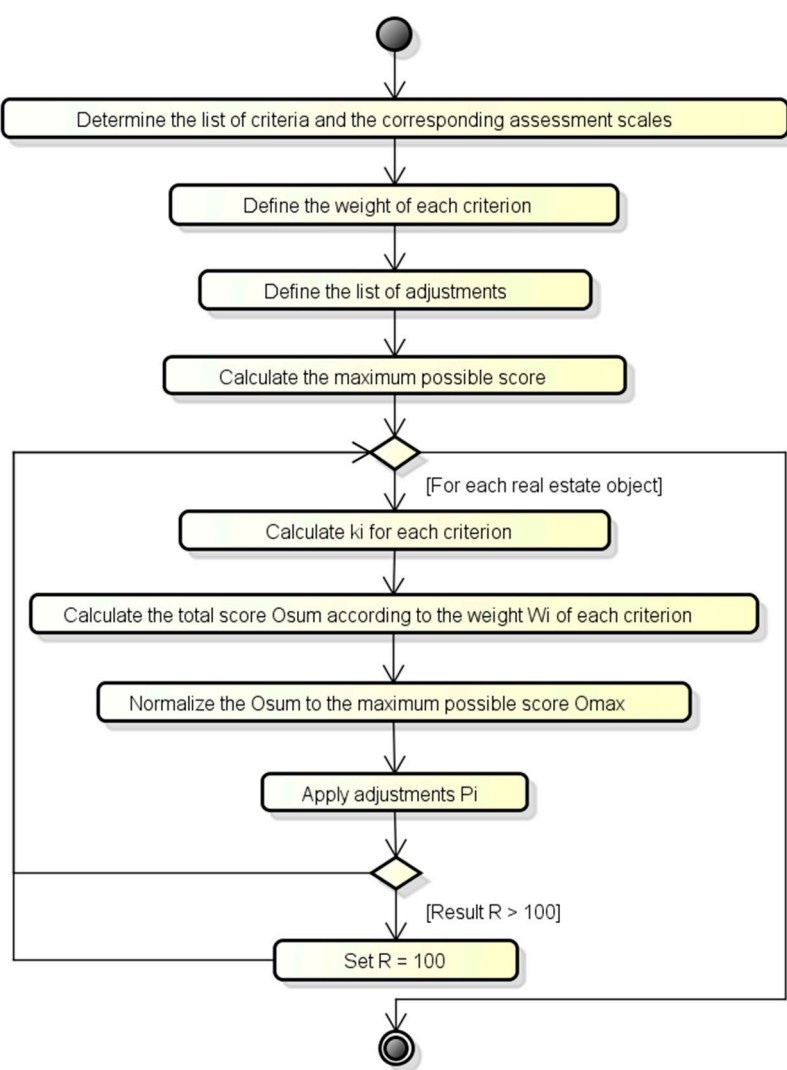

**Figure 5.** Algorithm for calculating attractiveness ratings for real estate objects.

The maximum possible total assessment of the object, taking into account the weight of each criterion, was calculated using the following formula [50]:

$$O_{\max} = \sum_{i=1}^{N} k_i \times O_{i\max},$$ (2)

where

$O_{\max}$—the maximum possible score for given scales and weights of each criterion;
$k_i$—weight of the $i$ criterion ($k_i = [0, 1], k_i \in R$);
$O_{i\max}$—integer maximum according to the evaluation scale of the $i$ criterion;
$N$—number of evaluation criteria.

The total ratings of objects for all criteria were calculated using the following formula [50]:

$$O_{sum} = \sum_{i=1}^{N} k_i \times O_i,$$ (3)

where

$O_{sum}$—total score for all criteria;
$k_i$—weight of the $i$ criterion ($k_i = [0, 1], k_i \in R$);

$O_i$—integer score for the $i$ criterion in points;
$N$—number of evaluation criteria.

The ratings were normalized according to the following formula [45]:

$$R1 = \frac{O_{sum}}{O_{\max}} \times 100\%, \tag{4}$$

where

$R1$—intermediate value of the overall score;
$O_{sum}$—total score for all criteria;
$O_{\max}$—the maximum possible score for given scales and weights of each criterion.

The adjustment mechanism was introduced to take into account parameters, the impact of which on the final assessment of consumer attractiveness is objectively known and does not depend on user preferences. This mechanism is similar to the mechanism of the same name, which is used in the market assessment of the value of real estate [54]. The principle of its operation is to use correction factors that numerically express how much the consumer attractiveness of an object with one value of a particular characteristic will differ up or down from the attractiveness of an identical object with a different value of the same characteristic, through the ratio of the assessment values of the attractiveness of these two objects.

The final values of assessments of consumer attractiveness of objects were calculated using the following formula:

$$R = R1 \times \prod_{i=1}^{M} kp_i, \tag{5}$$

where

$R$—final assessment of consumer attractiveness;
$R1$—intermediate evaluation value;
$kp_i$—$i$ correction factor ($kp_i \in R$);
$M$—number of characteristics taken into account through amendments for the type of object being evaluated.

This value was taken equal to 100% in cases where the final value of the assessment of consumer attractiveness $R$ as a result of the calculation turned out to be more than 100%.

The algorithm was developed without reference to a specific data structure and the list of evaluated characteristics. Lists of characteristics-criteria for evaluation, data on evaluation scales and correction factors with the conditions for their application in the framework of the study were implemented dynamically using configuration files. As a result, it became possible to change the data structure and adapt the evaluation procedure without changing the algorithm itself.

The algorithms described above make it possible to calculate the assessment of the attractiveness of each object of the area being developed before and after the implementation of the IAD project. The implementation of the calculation requires the availability of up-to-date data on the current state of objects in the area being developed and a description of the proposed IAD options that specify the final state of the area.

The impact on the attractiveness of each object is proposed to be calculated as:

$$w_i = (RA_i - RB_i)/RB_i \times 100\%, \tag{6}$$

where

$w_i$—percentage delta indicator of change in the attractiveness of the $i$ object;
$RA_i$—assessment of the object attractiveness after the IAD project implementation;
$RB_i$—assessment of the object attractiveness before the IAD project implementation.

A negative value of the indicator $w_i$ will indicate a decrease in the attractiveness of the object from the initial level, a positive value will indicate an improvement in attractiveness. A zero value will indicate that no change in the attractiveness of the object from the initial level happened.

The integral attractiveness delta for the entire territory can be calculated by obtaining the values of the indicator $w_i$ for each significant object in the area being developed:

$$W = \sum_{i=1}^{N} w_i \Big/ N, \tag{7}$$

where

$W$—integral attractiveness delta, expressed as a percentage;
$N$—total number of objects being assessed.

Comparison of IAD projects according to the calculated indicators $W$ will allow to conclude which of them has the best effect on the final consumer attractiveness of the area being developed.

### 4.2. Testing the Developed Methodology

The developed algorithm was tested on a sample of apartments for sale in the city of Volgograd, Russia. The data sample for testing the proposed evaluation method was formed by manually selecting ads from the sites Avito [55] and IRR [56] for the city of Volgograd.

It was necessary to take into account that some user ads can be unreliable. For example, realtors and real estate agencies can publish deliberately fake ads describing high quality objects which do not exist at all in order to lure customers. The sites where user ads are published can also be a favorite place for various scammers and frauds. Such an unreliable data sources is a very common thing in the sale or rental of residential real estate as, for example, apartments.

It was decided that such ads should be identified by filtering on the basis of a set of formal signs. Later this fact can be taken into account during attractiveness assessment of described objects, for example, by lowering the attractiveness level of objects from such "suspicious" ads.

The following formal signs of a 'suspicious' user ad were suggested to exclude potentially false ads from the selection:

- the ad contains known agency (realtor) or fraud phone number;
- the ad and five more ads contain the same phone number;
- the same phone number from the ad is mentioned in ads from different sellers;
- two or more ads describing different objects contain one or more same pictures (photos).

The perceptual hash algorithm was used as a solution of image comparing task in the framework of this study [57].

An algorithm of ads filtration is presented below (see Figure 6). This algorithm was implemented as a python script for automatic execution.

A testing sample of 100 two-room apartments offered for sale in the Dzerzhinsky District of the city of Volgograd was formed as a result of applying the developed script. Random objects were included in the sample with the presence of two rooms as the only common feature. Other signs were considered as the basis for the assessment, but their average or limit values were not decisive for the course of the study.

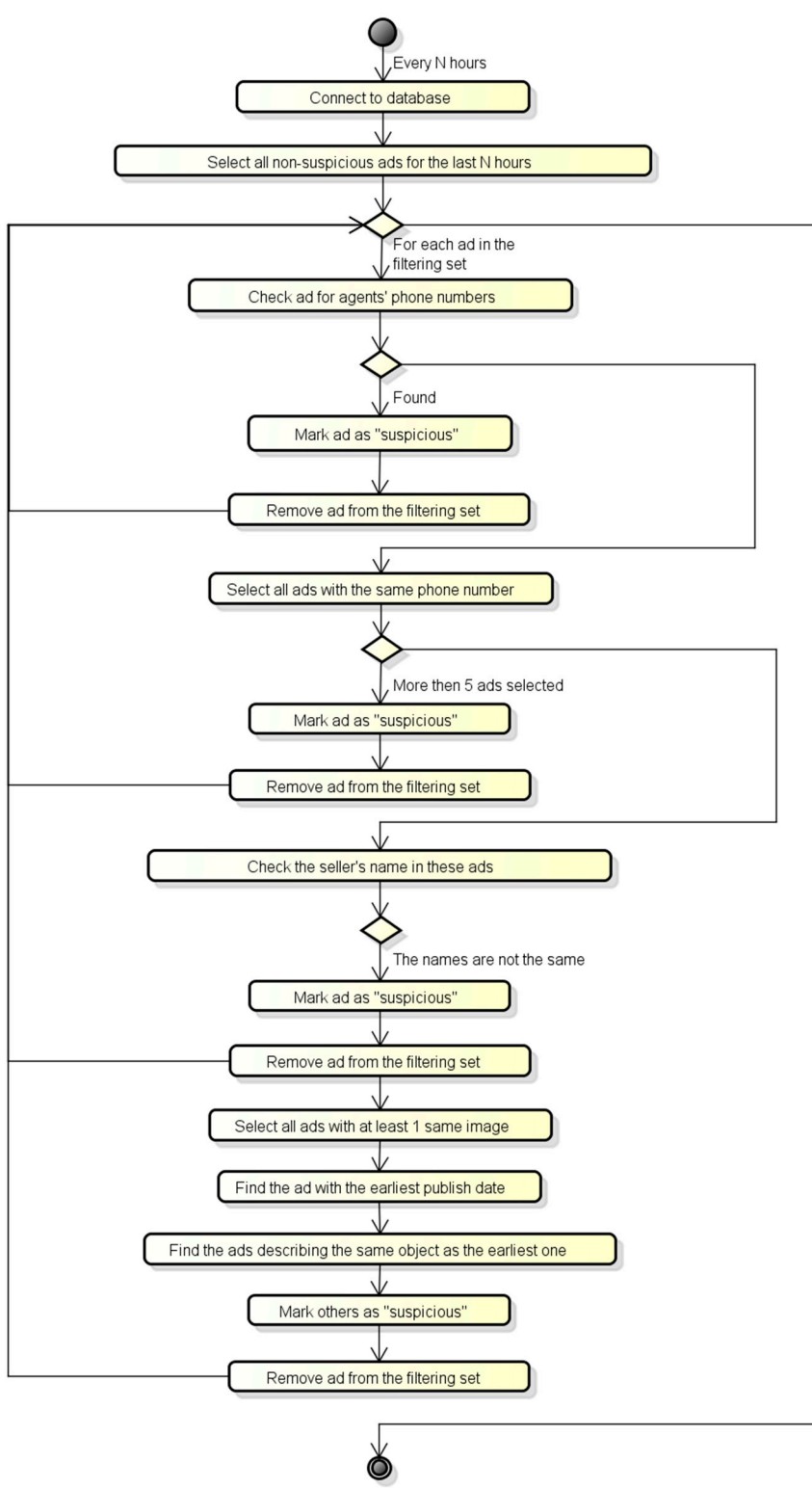

**Figure 6.** Algorithm of ads filtration.

The valuation of real estate objects was carried out according to the following characteristics:

- area (compliance with the area standards for a family of two people);
- house age;
- the type of house (brick, panel, etc.);
- apartment condition;

- the presence of an elevator;
- infrastructural provision.

A fragment of the test sample and the results of the attractiveness assessment algorithm are presented in Table 1 [58].

**Table 1.** Sample of apartments for the experiment and its results (fragment).

| Characteristic | Apartment 1 | Apartment 2 | Apartment 3 | Apartment 100 |
|---|---|---|---|---|
| Area | 49.6 | 66 | 68 | 62 |
| Elevators | 1 | 1 | 2 | 1 |
| Repair Status | – | cosmetic | European-quality | – |
| Construction Year | 1963 | 2009 | 1983 | 2010 |
| House Type | panel | brick | panel | monolithic |
| Pharmacy | 8 pcs, 270 m | 8 pcs, 550 m | 5 pcs, 160 m | 1 pcs, 1000 m |
| Grocery Store | 12 pcs, 130 m | 11 pcs, 400 m | 8 pcs, 450 m | 6 pcs, 450 m |
| Public Transport | 8 pcs, 500 m | 9 pcs, 260 m | 10 pcs, 240 m | 2 pcs, 850 m |
| Education | 12 pcs, 450 m | 12 pcs, 450 m | 14 pcs, 270 m | 1 pcs, 1000 m |
| Shops | 1 pcs, 700 m | 0 | 7 pcs, 1025 m | 3 pcs, 670 m |
| Healthcare | 1 pcs, 1000 m | 1 pcs, 600 m | 1 pcs, 108 m | 1 pcs, 600 m |
| Sport | 1 pcs, 620 m | 1 pcs, 320 m | 1 pcs, 548 m | 1 pcs, 2500 m |
| Attractiveness rating from 0 to 100 | 57 | 74 | 64 | 77 |

The studied sample of objects was sorted in ascending order of their cost. The dynamics of the obtained assessments of attractiveness was compared with the dynamics of market prices in the sample (Figure 7). Fluctuations in attractiveness ratings can be explained by the lack of a clear correlation between the two values. However, similar dynamics between the attractiveness of the object and the market price of the object allow to conclude that the developed approach is adequate.

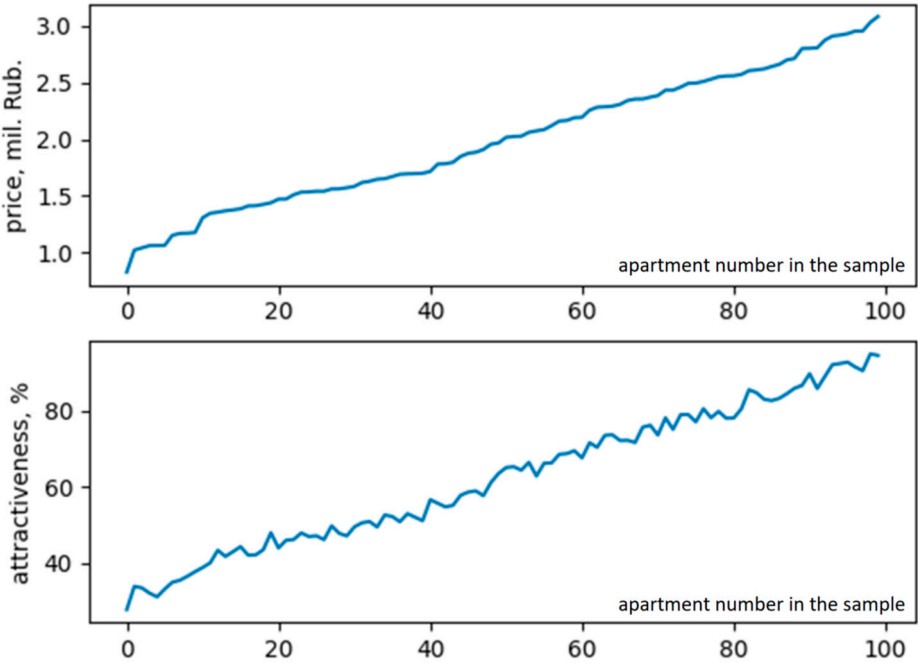

**Figure 7.** Comparison of price dynamics and attractiveness ratings.

A non-built-up area of 1.06 km$^2$ in the same Dzerzhinsky District of the city of Volgograd was selected for further experiment. The location of the site selected for the experiment on the map of the city of Volgograd is shown in Figure 8.

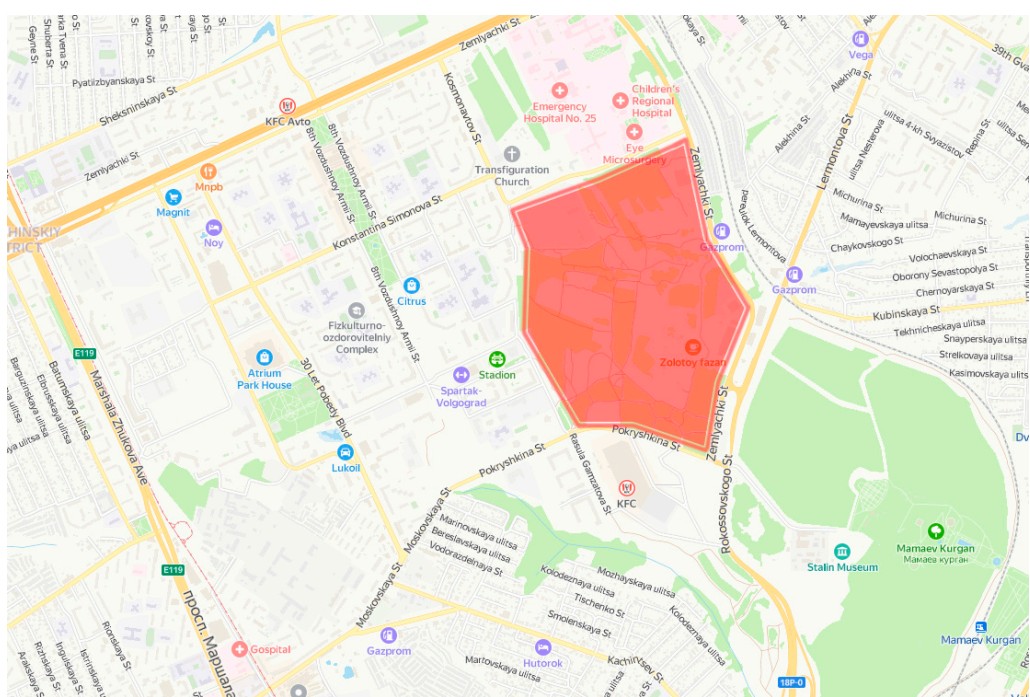

**Figure 8.** Non-built-up area, Volgograd (highlighted in red).

The city practically does not use this part of the territory. A mix of fading forest plantation with wild steppe plain and rare instances of commercial real estate (for example, there is a restaurant and an electronics store) is currently located on this site. Although the area may have some ecological significance for nearby urban areas and their inhabitants [59,60].

The purpose of the experiment was to test the proposed decision support method in terms of choosing from three options for developing the area. It is proposed to trace the changes in the attractiveness of individual objects and the integral indicator of the attractiveness of the entire study area of the city with the possible implementation of these three options within the existing sample.

Three development options for the selected area were modeled:

1.  complete residential development with associated commercial real estate (shops, pharmacies, etc.);
2.  the full development of commercial real estate (a business center with related infrastructure);
3.  the improvement of the territory with the preservation of the ecological role (replacement of the forest plantation with a park zone; use the remaining area for commercial real estate).

Previously collected data, on which real estate attractiveness was assessed, has been adjusted to reflect the state of the area and the selected site after the potential implementation of each of the proposed development projects. Next, a re-evaluation of objects was carried out for each possible project. The integral indicators of the attractiveness of the Dzerzhinsky District territory after the implementation of each project, as well as the integral indicator of the attractiveness of the district territory with the undeveloped area in its current state, were calculated upon completion of the assessment. The integral indicators of the attractiveness of the territory and the delta indicators of attractiveness obtained as a result of the experiment are presented in Table 2.

**Table 2.** Experiment results.

| Development Option | Overall Attractiveness | Attractiveness Delta |
|:---:|:---:|:---:|
| No Development | 59.76 | – |
| Residential Development | 59.94 | 0.18 |
| Commercial Development | 60.03 | 0.27 |
| Landscaping | 60.44 | 0.68 |

### 4.3. Interpretation and Discussion of the Obtained Results

The possibility of using machine learning methods to assess the consumer attractiveness of the territory was considered. However, there are no datasets or data sources for objects previously assessed in this way. The presence of a clear relationship between the attractiveness of the object (its general quality) and the market price of this object is determined. But this dependence does not have an exact mathematical expression. There is no formula or algorithm for converting a market price into a degree of attractiveness.

Therefore, the data about real estate market prices was deemed unsuitable for solving research task of this study. However, the ideas behind the market valuation of real estate, as well as information from the real estate appraiser's handbook [54], gave some guidance on how to evaluate consumer attractiveness of real estate objects.

The key aspect is to properly configure and set up the system implementing the suggested assessment approach, including:

- write context-free grammars for Tomita to correctly discover and extract the fact fields needed;
- predefine the list of the assessment criteria (most probably a separate list for each type of real estate to be assessed);
- set up the data necessary to calculate the criteria weights (which means involving experts in the field of real estate appraisal to assess the significance of each of the criteria);
- validate the data provided by experts etc.

The need to collect information from real estate ad sites in automatic, rather than manual mode, as was done for the test sample, is inevitable in the future. While this process comes with its own set of complexities and challenges to solve [61].

It is also necessary to perform filtration on information about real estate, since among the ads from thematic sites there may be ones that are deliberately fake.

Still the results presented in Table 1 and Figure 7 show that the price and the grade of attractiveness indeed share the similar dynamics. The noted correlation allows to conclude that the developed approach is adequate to the task and corresponds to the real world.

The increase in the attractiveness of the area during the experiment, presented in Table 2, cannot be called significant, since it is less than 1%. However, this situation is explained by the peculiarities of the experiment in the framework of this study.

The attractiveness of the entire district territory was assessed, and the proposed development changes were of a local nature and potentially affected the attractiveness of only individual objects located next to the area being developed. The calculation for the practical implementation of the proposed method should be carried out within the boundaries of smaller areas and only objects close to the developed area should be considered.

The development option with landscaping should be considered the best according to the proposed decision support method. Neighboring residential development with the current configuration of attractiveness calculations does not affect the attractiveness of neighboring objects. Therefore, the first development option increased the overall attractiveness only due to additional infrastructure facilities. The option with commercial real estate (business center) did not show much better results for the same reason. The landscaping option showed the best score due to the fact that there are no park areas around the study area.

This allows to conclude that the developed approach can indeed be used to assess consumer attractiveness of the whole city territory and then to compare IAD projects based on the total attractiveness delta of the areas under development. However, the amount of work applied to the experiment preparation suggests that, in addition to the approach proposed, the tools for filtering and validating data are essentially necessary.

## 5. Conclusions

This study attempts to look at one of the UN's key sustainable development goals through the prism of diverse human needs. Cities that are comfortable for life are not a whim, but a vital necessity. Today, it is already fully possible to speak about the civilization of the townspeople on our planet. The organization of sustainable development of urban areas has experienced several iterations in understanding the problems to be solved and approaches to their solution over the past almost two centuries. The current iteration is marked by the implementation of projects for the integrated areas development. Finally, this study compares the objectives of the IAD and sustainable development.

It Is proposed to implement a decision support system to achieve the desired development indicators. The assessment of the quality of the area being developed from the point of view of a person, as a consumer of its benefits, is proposed to be used as a key assessed indicator of the area development.

The methodology for assessing the area consumer attractiveness has been developed to calculate the proposed indicator. The methodology implements an approach based on the assessment of the entire set of real estate objects in the study area. Methods for collecting, filtering and processing information about real estate objects presented in open sources in natural language are described in the article. Algorithms for assessing the infrastructural provision of real estate objects and calculating estimates of their attractiveness are given. The results of testing the proposed methodology based on a prepared sample of data on apartments in the city of Volgograd, Russia, are presented in the article.

A comparison of the dynamics of the obtained assessments of consumer attractiveness with the dynamics of market prices in the sample is carried out to assess the adequacy of the results obtained. Attractiveness ratings under various IAD options for an urban district are presented.

It should be noted that this study was originally based on the data of sellers and landlords of real estate. At the same time, the final list of factors characterizing the quality of real estate objects was formed on the basis of an analysis of the real estate market in different countries of the world. Nevertheless, in future studies, it is planned to investigate the specifics of the regional aspect and socio-psychological factors that characterize the difference in perceptions of the quality of sellers/buyers of real estate [62].

In addition, high expectations regarding future research are associated with the development of an approach to solving the inverse problem: search for the optimal option for the integrated area development, taking into account the restrictions. Within the framework of this approach, a certain level of assessment of the user attractiveness of the set of objects of the developed territory will be considered as one of the target indicators for achieving the expected results. The tools of the developed methodology are supposed to be used to prepare data for the assessment of objects and infrastructure adjacent to the prospective development zones under study. The methodology for assessing the provision of the area with infrastructure needs to be improved for this.

**Author Contributions:** Conceptualization, D.P.; methodology, D.P. and I.Z.; software, I.Z. and A.G.; validation, D.P., I.Z. and A.F.; formal analysis, O.S. and A.G.; investigation, I.Z. and A.G.; resources, D.P. and A.F.; data curation, I.Z. and O.S.; writing—original draft preparation, I.Z., D.P. and O.S.; writing—review and editing, D.P. and A.F.; visualization, I.Z.; supervision, D.P.; project administration, D.P.; funding acquisition, D.P. and A.F. All authors have read and agreed to the published version of the manuscript.

**Funding:** The study has been supported by the grant from the Russian Science Foundation (RSF) and the Administration of the Volgograd Oblast (Russia) No. 22-11-20024, https://rscf.ru/en/project/22-11-20024/ (accessed on 16 October 2022). The results of Section 2 were obtained within the RSF grant project No. 20-71-10087.

**Institutional Review Board Statement:** Not applicable.

**Informed Consent Statement:** Not applicable.

**Data Availability Statement:** Not applicable.

**Conflicts of Interest:** The authors declare no conflict of interest.

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
