# Peer review of "Effective Implementation of Integrated Area Development Based on Consumer Attractiveness Assessment"

_sustainability, doi:10.3390/su142316239_

Round 1

Reviewer 1 Report

  1. The statement “The following 78 characteristics of the territory can be used as indicators:
    • transport accessibility
    • provision of social infrastructure
    • consumer attractiveness of the real estate, etc. 

Each of these characteristics needs to be briefly discussed within the introduction scope. 

  1. From the section introduction and background, this manuscript has not discussed the problem statement of this research. In other words, what is this research all about? What are the research questions that need to be answered? It needs some clarification. I can only find it in lines 197-204. It needs more discussions
  2. Since the case study was in Volgograd, Russia, it needs some background explanations in the introduction and the background sections. It is significant for readers to know some background of city regeneration of the city. 
  3. Section 3.2 is more suitable to put under the findings section
  4. In the discussion section, I think line 547-565 is unnecessary unless it discusses in the methodology section.
  5. The methodology section is more interesting than the findings section. The findings section is far more significant in knowing the impact of this study on the planning area.
  6. Therefore, the authors should rearrange the sequence of sections of this research. 
  7. There is a need to put the conclusion part in this research.

Author Response

The authors are grateful to the reviewer for informative comments, which made it possible to make the necessary editing to improve the quality of perception of the study. Please see the attachment with the corrected version of the article

  1. The statement “The following 78 characteristics of the territory can be used as indicators:

- transport accessibility

- provision of social infrastructure

- consumer attractiveness of the real estate, etc. 

Each of these characteristics needs to be briefly discussed within the introduction scope. 

Several characteristics are given in the text to provide a rough idea of the possible options. Now each of the characteristics has been given a brief summary in the text of the introduction.

  1. From the section introduction and background, this manuscript has not discussed the problem statement of this research. In other words, what is this research all about? What are the research questions that need to be answered? It needs some clarification. I can only find it in lines 197-204. It needs more discussions

An extended discussion was added at the end of the background, outlining aspects of the research problem statement based on the review and objectives described in the introduction and background text.

  1. Since the case study was in Volgograd, Russia, it needs some background explanations in the introduction and the background sections. It is significant for readers to know some background of city regeneration of the city. 

Appropriate related explanations have been added in the introduction and background, describing the prerequisites and relevance for conducting the presented study using the example of the city of Volgograd.

  1. Section 3.2 is more suitable to put under the findings section

We thank the reviewer for the suggestion, however, in this subsection, we describe the methods for obtaining materials on which the conclusions are based.

  1. In the discussion section, I think line 547-565 is unnecessary unless it discusses in the methodology section.

We consider it expedient to leave these lines. These lines briefly characterize the meaning of the technical and methodological problem of a purely cost assessment of the significance of the components of the territory and the composition of the elements of the proposed methodology, designed to become an alternative to the price approach, taking into account consumer attractiveness.

  1. The methodology section is more interesting than the findings section. The findings section is far more significant in knowing the impact of this study on the planning area.

We thank the reviewer for the assessment and suggestion. As far as possible, they took it into account when finalizing the material.

  1. Therefore, the authors should rearrange the sequence of sections of this research. 

With respect for the reviewer's suggestion, we revised the division of the material into structural components (this applies to paragraphs 4, 6 and 7 of the review). The section of the conceptual part with a general presentation of the methodology was separated. Sections of the results of the implementation of the methodology and discussions about the achievements were formalized as a single semantic component of the article.

  1. There is a need to put the conclusion part in this research.

A conclusion has been added with a brief description of the results obtained and future research.

Regards,
the team of authors

Reviewer 2 Report

The topic presented is very pertinent and I enjoyed reading it. However, in order to improve the article I present you my suggestions.

Abstract

The abstract is too big. According to the Instructions for Authors, the abstract should be a total of about 200 words maximum. The abstract should refer to the main results and conclusions. In this regard, the results and conclusion in the abstract should be clarified (“The results of testing the proposed methodology…” at the end of the article).

Introduction

Further, clarify what is in lines 68 to 71 on page 2. Are these research questions?

Which author supports the use of the indicators presented at the end of the introduction (page 2 lines 80, 81 and 82)?

Note that the objectives need to be well defined and it seems that the objectives they put in the abstract are not the same as the ones in the introduction.

Background, Methods and Results

·         In Figure 1 a correlation model is presented, but being a correlation model the representation should be with double arrows. In the case of the model in figure 1, the arrows are unidirectional, which means that, for example, the variable Economic Sphere Development influences the variable Integrated Area Development. Beware of the sentence on page 5 (lines 206 and 207) it seems that the model was obtained from the analyzed results and not theoretically.

·         In Figure 2 they get into a contradiction because in figure 1 the Sustainable Area Development variable was a dependent variable and in figure 2 it is already independent. Revise the models presented to avoid this contradiction!

·         In defining the indicator attractiveness what literature did they base it on? They should specify at the bottom of page 6.

·         In the flowchart in figure 4 an R appears, they should specify what it means, because then it appears in the formulas to calculate ki.

·         You are shown several formulas I_i, O_max, O_sum,... where do they enter in the algorithms represented by the flowcharts? You should put the letters used in the various formulas on the flowcharts, for example in parentheses, so that you understand where the algorithm uses them.

·         In formulas (4) and (6) you must remove the % symbol because the value obtained will be in per cent.

·         At the top of page 12, they talk about w_i > 0 and w_i < 0 and should add the case where w_i =0, where no change in attractiveness occurs.

·         It would be important to make a short text to characterize the sample made up of the 100 apartments, where you indicate the average area of the apartments (with the respective standard deviation), most of them have x elevators, etc. .... and also some descriptive measures to characterize the price and attractiveness rating (mean, SD, median, min, max, Sk, Ku).

·         In figure 7, you should specify what is represented on the abscissa axis. I think a scatter plot would be better to analyze the correlation between attractiveness and the price of the apartments, to conclude that the higher the value of the attractiveness indicator, the higher the price. If the correlation is high, you can also apply a linear regression to estimate the value of attractiveness as a function of price.

·         Add a section with the conclusions obtained in the study and its implications, and also talk about the limitations and future research.

·         I suggest reading the following article on determining the facts of the choice of apartments:

Santos, E., Tavares, F., Tavares, V., & Ratten, V. (2021). Comparative analysis of the importance of determining factors in the choice and sale of apartments. Sustainability, 13(16), 8731.

Good luck in your future researches

Author Response

  1. The topic presented is very pertinent and I enjoyed reading it. However, in order to improve the article I present you my suggestions.

We thank the reviewer for their attention and valuable comments, which allowed us to clarify and improve the description of the presented study. Please see the attachment with the corrected version of the article.

Abstract

  1. The abstract is too big. According to the Instructions for Authors, the abstract should be a total of about 200 words maximum. The abstract should refer to the main results and conclusions. In this regard, the results and conclusion in the abstract should be clarified (“The results of testing the proposed methodology…” at the end of the article).

The abstract has been rewritten to include the stated findings and to comply with the scoping requirements.

Introduction

  1. Further, clarify what is in lines 68 to 71 on page 2. Are these research questions?

This list provides examples of issues faced by a decision maker in integrated urban planning. This is indicated in the text before the specified enumeration.

  1. Which author supports the use of the indicators presented at the end of the introduction (page 2 lines 80, 81 and 82)?

Unfortunately, the question is not completely clear, but we have given a detailed description in the text of the introduction to clarify the content of the specified list for each of the characteristics given as an example.

  1. Note that the objectives need to be well defined and it seems that the objectives they put in the abstract are not the same as the ones in the introduction.

The objectives of the study were structured in the introduction. When adjusting the abstract, the wording of the research goal was clarified to clearly match the description given in the introduction.

Background, Methods and Results

  1. In Figure 1 a correlation model is presented, but being a correlation model the representation should be with double arrows. In the case of the model in figure 1, the arrows are unidirectional, which means that, for example, the variable Economic Sphere Development influences the variable Integrated Area Development. Beware of the sentence on page 5 (lines 206 and 207) it seems that the model was obtained from the analyzed results and not theoretically.

We agree with the proposed adjustments. Figure 1 was changed in accordance with the recommendation, double arrows were added. The indicated lines have been corrected, since the model was actually obtained based on the results of the analysis.

  1. In Figure 2 they get into a contradiction because in figure 1 the Sustainable Area Development variable was a dependent variable and in figure 2 it is already independent. Revise the models presented to avoid this contradiction!

Figures 1 and 2 consider various aspects of the topic under study, therefore they do not have a direct relationship (they are not a decomposition of each other). Therefore, the authors think that the presented schemes do not contradict each other.

  1. In defining the indicator attractiveness what literature did they base it on? They should specify at the bottom of page 6.

The choice of the composition of factors of consumer attractiveness is the product of the author's and expert work.

  1. In the flowchart in figure 4 an R appears, they should specify what it means, because then it appears in the formulas to calculate ki.

Thanks for pointing out the confusion. Removed the letter R from the flowchart, because it does not carry a key meaning there and was used in the separate development of the algorithm.

  1. You are shown several formulas I_i, O_max, O_sum,... where do they enter in the algorithms represented by the flowcharts? You should put the letters used in the various formulas on the flowcharts, for example in parentheses, so that you understand where the algorithm uses them.

Thanks for the recommendation. Marked the indicated symbols on the flowcharts (Figures 4 and 5).

  1. In formulas (4) and (6) you must remove the % symbol because the value obtained will be in per cent.

We believe that the % symbol cannot be removed from these formulas, since otherwise, the resulting values will be in fractions multiplied by 100.

  1. At the top of page 12, they talk about w_i > 0 and w_i < 0 and should add the case where w_i =0, where no change in attractiveness occurs.

Thanks for the advice. An appropriate explanation has been added to the text.

  1. It would be important to make a short text to characterize the sample made up of the 100 apartments, where you indicate the average area of the apartments (with the respective standard deviation), most of them have x elevators, etc. .... and also some descriptive measures to characterize the price and attractiveness rating (mean, SD, median, min, max, Sk, Ku).

An explanation was given for the sample used and the features of the objects used. The sample is random and the identification of some common features for its characteristics is not significant for the results of the study.

  1. In figure 7, you should specify what is represented on the abscissa axis. I think a scatter plot would be better to analyze the correlation between attractiveness and the price of the apartments, to conclude that the higher the value of the attractiveness indicator, the higher the price. If the correlation is high, you can also apply a linear regression to estimate the value of attractiveness as a function of price.

The abscissa axis designation has been added. We did not merge the charts and display the attractiveness as a scatter plot due to the fact that the values are visually very close to the price values. The current version of the graphs clearly displays the correlation of values.

We would like to thank the reviewer for the idea of using linear regression, which will be used in future studies.

  1. Add a section with the conclusions obtained in the study and its implications, and also talk about the limitations and future research.

A conclusions section was added, including a description of further research.

  1. I suggest reading the following article on determining the facts of the choice of apartments:

Santos, E., Tavares, F., Tavares, V., & Ratten, V. (2021). Comparative analysis of the importance of determining factors in the choice and sale of apartments. Sustainability, 13(16), 8731.

Thanks for the suggested material. The article touches on a really interesting aspect of the behavior of real estate market participants, the impact of which should be studied separately in the context of our research. We supplemented this idea in the conclusion in promising work.

  1. Good luck in your future researches

We thank the reviewer for good wishes and very attentive attitude to the reviewed material.

Regards,
the team of authors

Round 2

Reviewer 1 Report

All comments and revision have been highlighted in the revised version

Reviewer 2 Report

Many successes to continue with future investigations.